# Monopath DAGs: Structuring Patient Trajectories from Clinical Case Reports

## Abstract

High-quality datasets capturing rare diseases, atypical responses, and complex care pathways are critically needed in clinical machine learning. While electronic health records (EHRs) remain the dominant data source, they are constrained by institutional silos, privacy regulations, and the inherent scarcity of many clinically significant scenarios. Narrative case reports offer a complementary source: publicly available and often focused on diagnostically or therapeutically challenging cases. Yet their unstructured format limits reuse for modeling and data generation. We present a modular framework that transforms free-text case reports into *Monopath Directed Acyclic Graphs (DAGs)* —structured representations of patient trajectories that are both temporally ordered and semantically grounded. DAGs are a natural fit for modeling clinical narratives as they encode time-ordered clinical states and transitions, supporting branching and causal reasoning. We apply the pipeline to a curated corpus of 485 lung cancer case reports. Graph fidelity is supported both by automated metrics (ClinicalBERT BERTScore, $F_1 = 0.798 \pm 0.051$) and by direct clinical assessment, with practicing physicians rating event order and content positively. Compared to free-text vignettes, DAG embeddings yield higher Calinski–Harabasz clustering scores in raw space (110.5 vs. 41.9) and after PCA/UMAP (157% and 69% relative gains). In a clinician evaluation, graph-conditioned synthetic narratives are preferred in 62% of 106 comparisons and scored higher on timeline validity and decision support. In addition, we demonstrate applicability beyond lung cancer by applying the framework to four rare diseases across body systems, observing consistent clustering gains. Pending large-scale validation, these results highlight the promise of Monopath DAGs to serve as reusable, clinically grounded templates for patient similarity, augmentation, and controlled narrative generation. We release all graphs, schema, and code.

## 1 Introduction

High-quality datasets capturing rare diseases, atypical responses, and complex care pathways are critically needed in clinical machine learning. Electronic health records (EHRs)—the dominant data source—are limited by institutional fragmentation, privacy protections, and the rarity of many scenarios. Even large public datasets such as MIMIC-III (Johnson et al. (2016)), MIMIC-IV (Johnson et al. (2023)), and eICU (Pollard et al. (2018)) disproportionately represent common conditions and inpatient encounters, with limited temporal continuity, incomplete outcome tracking, and underrepresentation of multimodal interventions. Moreover, EHRs are structured primarily for billing and documentation, making it difficult to extract coherent longitudinal trajectories or rare clinical events. As a result, they seldom capture the complexity, edge cases, or dependencies needed to advance model development and mirror real clinical environments.

Narrative case reports offer a promising but underutilized alternative. These peer-reviewed, de-identified publications are authored by clinicians to illustrate clinical reasoning, temporal evolution, and cause-effect relationships that are rarely captured in EHRs. They are particularly valuable for surfacing underrepresented scenarios, such as rare, complex, or diagnostically challenging presentations. However, their unstructured free-text format poses significant challenges for use. Temporal relationships, decision points, and state transitions are often implicit, context-dependent, and described in domain-specific language. As a result, despite their availability in large numbers, case reports have remained largely inaccessible for data-driven modeling and large-scale analysis.

Beyond structure, it is critical to provide machine learning systems with datasets that align with the processes by which clinicians reason in the real world. In practice, medical decision making involves chaining causes to effects under uncertainty, relying on tacit knowledge that is often missing in explicit labels and does not appear as token overlaps. Yet, most existing datasets and benchmarks reward models for surface-level recall or single-label accuracy. This systematically overlooks signals that govern safe and effective care, including temporal orientation, causal coherence, and decision specificity. The result are models that may have correct outputs in most scenarios, but may be misaligned with clinical reasoning, leading them to be potential unreliable in more complex or rare scenarios.

We introduce a data generation framework that transforms clinical case reports into structured representations of patient trajectories. The core output of this pipeline is a Monopath Directed Acyclic Graph (DAG), a graph-based format in which each patient journey is encoded as a single directed path through clinically meaningful states, driven by an event, observation, or intervention. Monopath DAGs are well suited to this task as they maintain temporal clarity and semantic precision, making them ideal for reasoning over longitudinal patient narratives. The framework integrates large language models (LLMs) with rule-based extraction and clinical ontology alignment to identify key entities, events, and transitions in the text. It then organizes this information into temporally grounded nodes and edges, enforcing acyclicity and preserving causal flow. We apply our pipeline to 485 curated lung cancer case reports and, to assess generalizability, further evaluate it on 200 rare disease cases, observing consistent improvements across tasks.

This work makes four key contributions:

- **A modular framework** that extracts temporally ordered, ontology-grounded DAGs from free-text case reports using LLMs, released with code and schema for reproducibility.
- **A structured dataset of 485 patient trajectory graphs** from curated **lung cancer case reports**, complemented by evaluations on **200 rare disease cases across four types**.
- **A clinically meaningful synthetic data generator** that perturbs extracted graphs to build synthetic trajectories aligned that prioritize fidelity.
- **An evaluation paradigm grounded in clinical decision-making**, assessed on five clinician-defined criteria, reflecting how clinicians build confidence in decisions to ensure that the dataset reflects clinical utility rather than surface accuracy.

## 2 RELATED WORK

**Clinical Narrative Information Extraction.** Prior work in clinical NLP has largely focused on structured sources such as clinical notes, discharge summaries, and radiology reports. Early systems (*e.g.*, cTAKES, MetaMap) combined rules and vocabularies for entity recognition and normalization. Subsequent methods included semi-supervised CRFs for temporal event detection in EMRs (Moharasar & Ho (2016)) and PatternCausality, an unsupervised dependency-based approach to extract cause–effect relations (Kabir et al. (2022)). With pretrained models, transformer-based methods such as ClinicalBERT (Alsentzer et al. (2019)), BioGPT (Luo et al. (2022)), and BlueBERT advanced entity and temporal extraction, while GraphTrex achieved state-of-the-art performance on i2b2 TempEval (Chaturvedi et al. (2025)). Case reports have also been targeted: GPT-3 has been used for extraction (Sciannameo et al. (2024)), and Zhou *et al.* developed CREATe for graph-based cardiovascular case report representations (Zhou et al. (2021)). Complementing this, PMC-Patients released 167,000 case report summaries for retrieval tasks (Zhao et al. (2022)), though the unstructured text limits temporal or causal modeling. In contrast, we transform full case narratives into structured, temporally ordered graphs, enabling reasoning, clustering, and simulation. More broadly, while most modeling still centers on structured EHR data, case reports demand methods that account for narrative complexity and rare or atypical presentations—motivating approaches that integrate statistical modeling with clinical knowledge to capture this richness in machine-readable form.

**Patient Trajectory Modeling.** Growing work has focused on structured EHRs and time-series data. Sequence-based models such as RNNs, Transformers, and temporal point processes have been applied to EHR sequences [*e.g.*, RETAIN (Choi et al. (2017)), BEHRT Li et al. (2020), Med-BERT (Rasmy et al. (2021))]. Graph-based methods have also emerged, treating visits or diagnoses as nodes; for example, GraphCare integrates knowledge graphs while modeling trajectories for clustering and prediction (Jiang et al. (2023)). Yet these approaches depend on structured inputs. Methods

for extracting trajectories directly from unstructured narrative case reports—essential for studying clinical progression beyond what EHRs capture—remain largely unexplored.

**Synthetic Patient Data Generation.** To address EHR data scarcity and privacy, synthetic generation has gained traction. Rule-based simulators like Synthea (Walonoski et al. (2018)) create lifelike trajectories using hardcoded progression logic, while ML-based approaches such as EHR-M-GAN (Li et al. (2023)) and EHR-Safe (Yoon et al. (2023)) leverage GANs and encoder–decoder models to synthesize longitudinal data with privacy safeguards. The HALO model (Theodorou et al. (2023)) applied hierarchical Transformers to generate disease code sequences, and GPT-style methods have been explored for discharge summary synthesis (Falis et al. (2024)). These techniques enable robust training without compromising privacy, but struggle to reproduce the rare, atypical trajectories and complex interventions documented in case reports—patterns that are difficult to simulate de novo.

## 3 MODULAR FRAMEWORK FOR PATIENT TRAJECTORY GRAPH EXTRACTION

We present a modular framework for converting unstructured clinical case report text into a structured representation of patient trajectories in the form of *Monopath Directed Acyclic Graphs (Monopath DAGs)*. A Monopath DAG is a specialized form of DAG in which each node has at most one incoming edge—reflecting a single dominant clinical pathway while maintaining temporal ordering and acyclic structure. Monopath DAGs offer a simplified yet expressive scaffold that captures the sequential progression of a patient's clinical states without ambiguity in causal flow.

### 3.1 TASK DEFINITION

We consider the task of representing each clinical case report $d_i \in \mathcal{D}$ as a directed acyclic graph (DAG), denoted $G_i = (V_i, E_i)$. In this formulation, nodes correspond to temporally ordered patient states, each annotated with free-text descriptions, structured attributes, and, when available, explicit timestamps. Directed edges capture transitions between states, labeled by their type and the clinical entities that trigger them. This graph-based representation provides a machine-readable abstraction of longitudinal patient trajectories, suitable for downstream computational modeling and analysis (see Appendix A.1 for the detailed formal definition).

### 3.2 DSPY-GUIDED PATIENT TRAJECTORY MONOPATH DAG GENERATION

We introduce a modular DSPy-driven pipeline (Figure 1) that converts clinical case reports $d_i \in \mathcal{D}$ into structured, temporally ordered patient trajectories $G_i = (V_i, E_i)$, where $V_i$ denotes clinical state nodes and $E_i$ directed transitions (Appendix C). Our system is designed to produce granular monopath DAGs with optional branching and Unified Medical Language System (UMLS)-grounded metadata.

**Preprocessing & Timeline Extraction.** Each case report is a PubMed Central UTF-8 HTML document with boilerplate removed. We segment the remaining narrative into 10-sentence blocks and pass them to a timeline generator $T_i = \texttt{PatientTimeline}(p_1, \ldots, p_k)$. After finding smaller open models (*e.g.*, Llama 3.1 8B/70B) insufficient for reliable node construction, we adopted Gemini 2.0 Flash ($T = 0.2$, $\sim$128k context) for more consistent performance and output fidelity.

**Node Construction.** The timeline $T_i$ is divided into 4-sentence blocks $c_j$, each processed by the DSPy program `NodeConstruct` to yield nodes $v_j = \{\texttt{node\_id}, \texttt{step\_index}, s_j, a_j, t_j\}$, where $s_j$ is node content, $a_j$ structured clinical data, and $t_j$ an optional timestamp. To prevent redundancy, we generate each node with access to prior memory $\{v_1, \ldots, v_{j-1}\}$. Nodes are merged by concatenating non-duplicate content and combining clinical data category-wise. We apply two DSPy signatures: `DecomposeToAtomicSentences`, which splits $s_j$ into atomic-level facts, and `NodeClinicalDataExtract`, which maps these atomic facts to ontology terms. Unmatched terms are preserved as free text, and merged nodes are reindexed deterministically by the `node\_id` and `step\_index`.

**Edge Construction.** Given the final node list $\{v_1, \ldots, v_n\}$, each adjacent pair $(v_i, v_{i+1})$ is passed to the `edgeConstruct` signature. An edge $e_{i \rightarrow i+1} = \{\texttt{edge\_id}, \texttt{branch\_flag}, \tau_{ij}, \phi_{ij}\}$ includes a transition description $\tau_{ij}$ and a structured event $\phi_{ij}$ with trigger type, domain, and ontology terms (*e.g.*, "`trigger\_type=medication\_change`"). Few-shot prompting (3 demonstrations)

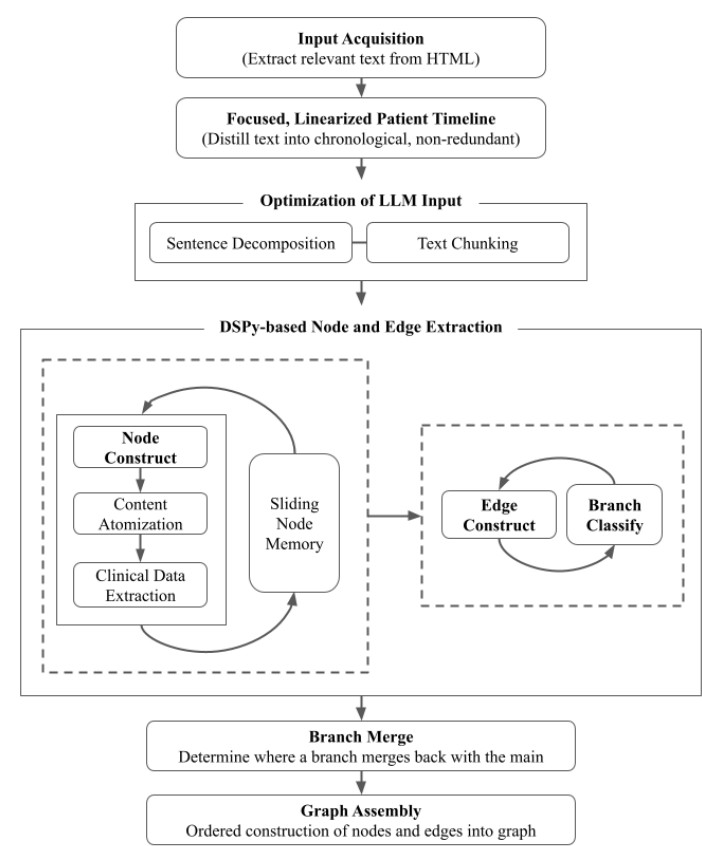

Figure 1: DAG generation pipeline for patient trajectory modeling.

enforces schema compliance and transition semantics. In practice, case reports are segmented into ~10-sentence blocks for timeline extraction and 4-sentence blocks for node construction, chosen heuristically to balance fidelity, granularity, and efficiency. These settings are not tuned on held-out data, ensuring generalizability.

**Branch Classification & Merge.** Branching edges—typically reflecting ephemeral complications—are detected using the `branchClassify` signature. For few-shot prompting, DSPy's BootstrapFewShot search is run on 85 labeled branch examples. The search consistently selected $k = 4$ demonstrations as yielding the highest dev-set exact-match accuracy. A sweep over $k = 2$–6 showed diminishing returns beyond four examples ($< 1\%$ improvement at $k = 5$–6), so we adopt $k = 4$ to balance performance with prompt length. Only the earliest edge with predicted `branch_bool=true` is marked as the branch origin. While merges are handled by topological rules (Appendix, Algorithm 2, C.5), for these experiments we linearize each graph by extracting its longest path to represent the patient sequence. Branching edges denote transient detours that ultimately rejoin the main trajectory.

**Graph Serialization & Metadata Tracking.** Each patient trajectory is represented as a JSON-based directed graph, where nodes capture temporally ordered clinical states and edges denote inferred transitions. Graphs follow a consistent schema with ontology-grounded labels and structured metadata (Appendix C). We enforce schema compliance during construction but defer full structural validation to support downstream evaluation on process.

### 3.3 DATASET OF LUNG CANCER CASE REPORTS

We curated a dataset of free full-text oncology case reports from PubMed, focusing on lung cancer. Using the Entrez API, we searched for case reports published from 2019 onward with open-access availability in PubMed Central (PMC). The query included a broad set of lung cancer–related

terms—lung cancer, lung carcinoma, non–small cell lung cancer (NSCLC), small cell lung cancer (SCLC), mesothelioma, pulmonary carcinoma, and bronchogenic carcinoma—matched in the title and abstract fields. From this search, we assembled a pool of 549 reports, targeting a dataset on the order of $5 \times 10^2$. This number represented a practical stopping point given time constraints and available clinician resources, and provides sufficient diversity and depth to evaluate the framework.

## 4 EVALUATION AND UTILITY

### 4.1 GRAPH CONSTRUCTION AND FIDELITY

To ensure the LLM- and rule-based transformation with ontology alignment faithfully captures the original clinical narrative, we implement a comprehensive fidelity evaluation framework combining automated and human-in-the-loop assessments.

We first reconstruct the input case narrative from the generated graph using DSPy-compatible Gemini-2.0-Flash. The model receives a payload containing the graph's nodes and edges and is prompted to generate a coherent textual summary that mirrors the structure and content of the original case. The reconstructed narrative $\hat{R}$ is compared to the original case report $R$ using BERTScore (Zhang et al. (2019)), which measures semantic similarity via contextualized embeddings from ClinicalBERT. Precision and recall are computed as $\text{Precision} = \frac{1}{|\hat{R}|} \sum_{\hat{r} \in \hat{R}} \max_{r \in R} \text{sim}(\hat{r}, r)$ and $\text{Recall} = \frac{1}{|R|} \sum_{r \in R} \max_{\hat{r} \in \hat{R}} \text{sim}(r, \hat{r})$, where sim denotes cosine similarity; F1 is the harmonic mean of these values.

Structural integrity of the patient trajectory graphs is ensured by implementing a validation routine that checks key topological properties. For each graph, we assess acyclic nature through depth-first search. We compute the number of weakly connected components to detect disconnected subgraphs, and extract summary statistics including node and edge counts, average in-degree, and graph density. These metrics provide essential quality control and enable downstream filtering of malformed or incomplete graphs. This ensures that the output graphs preserve logical flow and are suitable for downstream trajectory modeling. Clinical annotators perform a comparative assessment of the original case report and its corresponding graph. to evaluate fidelity and quality. (Appendix D.2)

### 4.2 CLUSTERING OF PATIENT TRAJECTORIES

We evaluate patient trajectory representations in two modalities: (1) derived Monopath DAGs, and (2) corresponding free-text cases. Both representations are embedded using a pretrained Clinical-BERT model to yield fixed-length vector embeddings. For graph-based representations, token-level embeddings from node text are aggregated using mean pooling. For text-based representations, the entire vignette is encoded directly. This approach enabled comparison of structural (graph-derived) versus linguistic (text-derived) patient representations, while controlling for embedding architecture and pretraining data.

To prepare high-dimensional embeddings for clustering, we apply two complementary dimensionality reduction methods: (i) PCA with whitening to de-correlate dimensions and normalize variances, and (ii) UMAP with default hyperparameters (15 neighbors, minimum distance = 0.1, Euclidean metric) as recommended by McInnes *et al.* (McInnes et al. (2020)). These widely adopted defaults provide a robust, reproducible baseline. We then apply K-means clustering to the reduced embeddings, evaluating cluster configurations with $k \in [2, 10]$. Clustering quality is assessed using the Calinski-Harabasz Index (CH) for both graph-based and text-based embeddings under three settings: raw, PCA-reduced, and UMAP-reduced. To examine clinical relevance, we incorporate structured metadata on binary metastasis status, computing the proportion of metastatic versus non-metastatic cases within each cluster and normalizing for cluster size to enable fair comparison.

### 4.3 SYNTHETIC TRAJECTORIES AND HUMAN CLINICAL EXPERT REVIEW

**Synthetic Generation.** For a random subset of our dataset, for each patient trajectory graph $G_i$ we create two narratives. Control cases are reconstructed from the original HTML report: the raw HTML is summarised into node-like snippets and re-combined into prose with fixed DSPy prompts emulating naive language model prompting. Sample cases are generated from the graph itself: we

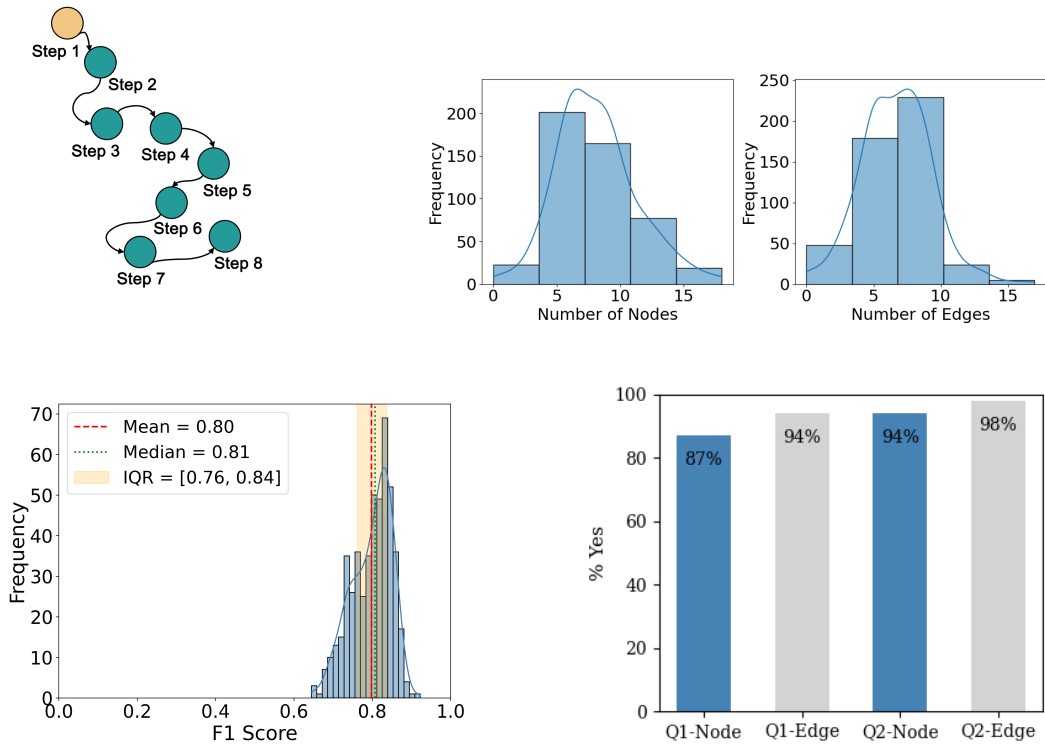

Figure 2: **Monopath DAG trajectory representations and evaluation.** (a) Schematic of a monolithic DAG representing a patient case, where nodes correspond to clinically meaningful events and directed edges capture temporal transitions. (b) Combined distribution of node and edge counts across all constructed graphs. (c) Histogram of BERTScore F1 scores evaluating similarity between reconstructed narratives (from graphs) and the original vignette texts. (d) Human evaluation comparing constructed graphs and corresponding original texts in terms of order (Q1) and accuracy (Q2).

take the first root-to-leaf path returned by a depth-first search (ignoring forward-jump edges), pass its nodes and edges to an `LLMReconstructor`, and obtain a graph-consistent narrative. Both pipelines use `gemini-2.0-flash` via DSPy. The resulting texts plus metadata (graph ID, model, node path) are saved for subsequent benchmarking.

> ### Clinician-centered evaluation criteria (Q1–Q5)
>
> - **Q1 (Clinical validity):** baseline check that case actions are medically sound.
> - **Q2 (Timeline validity):** assesses whether the case progression follows a clinically appropriate order, capturing temporal reasoning.
> - **Q3 (Clinically actionable):** measures whether the case is specific and granular enough to guide the next clinical decision.
> - **Q4 (Safely clinically actionable):** measures whether the information is sufficient to make a safe decision with confidence, reflecting decision certainty.
> - **Q5 (Appropriate language):** evaluates whether the narrative uses medical language appropriately, avoiding generic or ambiguous phrasing.

**Human Clinical Expert Review.** We conducted a single-blinded evaluation with four practicing physicians, who are shown a mix of real and synthetic patient graphs and asked to rate each on a 5-point Likert scale for realism and clinical plausibility. To move beyond correctness as the sole benchmark, we designed a clinician-centered framework that reflects the auxiliary principles guiding real-world decision-making. Five targeted questions (Q1–Q5), reviewed by medical experts, captured hidden reasoning signals—validity, temporal orientation, specificity, safety, and trust in

communication—that are absent from standard benchmarks but central to clinical impact. (See Appendix D.3 for the complete set of questions)

## 5 RESULTS AND DISCUSSION

From the initial set of 549 case reports, we remove non-English entries and case series, resulting in a curated dataset of 485 individual narratives. Semantic fidelity of the graph-derived reconstructions is evaluated using BERTScore, yielding an average BERTScore F1 of 0.798, median 0.807, and an interquartile range between [0.761, 0.836]. Scores span from 0.676 to 0.876, with a low ($\sigma \approx 0.051$), pointing to stable reconstruction quality across cases. The lower F1 scores observed point to a subset of challenging cases with lower alignment. These outliers may reflect failures in content preservation or structural reconstruction. Overall, the high average scores and low variance demonstrate that the model reliably reproduces the core clinical information and narrative structure across diverse cases. Precision and recall components of BERTScore are 0.790 and 0.807, respectively (Appendix B.1). On average, graphs contained $8 \pm 3$ nodes and $7 \pm 3$ edges, indicating clinically meaningful structural detail. Human evaluation of 568 nodes and edges further supports the structural consistency of the generated graphs. Node order is correct in 87% of cases, and 94% of nodes are deemed semantically accurate. For edges, correctness reaches 94%, and semantic validity approaches 98 %, reinforcing the effectiveness of the graph generation process in preserving temporal logic and clinical meaning (Figure 2b).

Clustering analyses across raw, PCA-reduced, and UMAP-reduced embedding spaces consistently favor graph-based representations over text. In the raw space, graphs attain a Calinski-Harabasz score of 110.5, significantly exceeding the 41.9 achieved by text-based embeddings. Dimensionality reduction preserves this advantage: PCA boosts the score to 207.9 for graphs (*vs.* 80.9 for text), while UMAP accentuates it further—489.2 compared to 289.4. The structured encoding of clinical events within graphs appears to enhance cluster separability, capturing nuanced variation across patient trajectories. When stratifying patients by metastasis status, graph-based clusters display sharper separation. As illustrated in Figure 3, the graph-based clustering (left) yields three distinct groups. Cluster graph-0 and graph-2 show strong enrichment for metastatic cases, with over 80% of instances labeled as metastatic, while graph-1 presented a more mixed composition with approximately 65% metastatic and 34% non-metastatic cases. In contrast, the text-based clustering (right) reveals four clusters with more variable metastatic distributions. Clusters text-0, text-1, and text-2 each contain over 70% metastatic cases, with text-1 being the most enriched ($\approx 85\%$). However, text-3 shows the lowest metastatic proportion ($\approx 63\%$) and the highest fraction of non-metastatic cases among text clusters. These findings indicate that graph-based embeddings more distinctly stratify patients by metastasis status compared to text-based representations. This suggests that structured, temporally grounded patient graphs may better capture disease progression signals relevant to clinical outcomes.

A blinded evaluation by four physicians (PGY-1 to PGY-3) compared control-generated and graph-conditioned synthetic cases across the five clinical criteria (Figure 4, Table 1). Results highlight that while clinical validity (Q1) remained stable across conditions, graphs yielded significant improve-

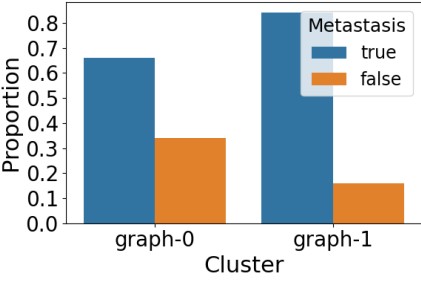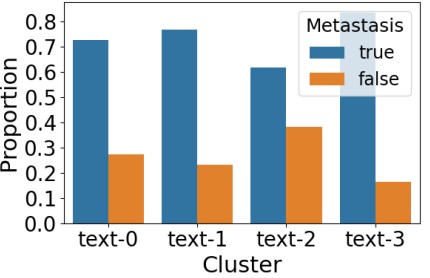

(a) Metastasis per cluster (Graph)    (b) Metastasis per cluster (Text)

Figure 3: **Cluster-level metastasis composition from graph- and text-based embeddings.** (a) Graph-based embeddings show distinct enrichment patterns by metastasis status across clusters. (b) Text-based embeddings exhibit similar trends, though with more heterogeneous distributions.

ments in timeline validity (Q2, $p = 0.003$), actionability (Q3, $p = 0.004$), and safe actionability (Q4, $p < 0.001$). These three dimensions are critical because they reflect the reasoning traces clinicians need: chronological progression, specificity for decision support, and confidence in safety. Language appropriateness (Q5) trended positive (p = 0.051), suggesting graphs may reduce generic or buzzword-heavy phrasing. Taken together, these results demonstrate that Monopath DAGs do not simply preserve correctness but enhance the qualities that make cases useful for clinical decision-making. Although the binary evaluation schema (Q1: order, Q2: accuracy) provides only a coarse-grained view of graph fidelity, it is designed to complement automated semantic metrics such as BERTScore with a minimal human check that could be performed within the constraints of clinician time. This design emphasized inter-reviewer agreement and feasibility, but it does not capture node informativeness or the clinical validity of edges on its own. We consider this a key limitation of the current evaluation and plan to extend our framework with more fine-grained protocols, including explicit node-level assessments and validation of transition plausibility. As noted in Section 4.3, clinician evaluation compared synthetic narratives with control narratives on textual criteria; physicians did not review graphs directly or express preferences between graph and text formats. In this study, graphs are used as structured intermediates to support generation and retrieval, rather than as interfaces intended for direct clinical decision-making.

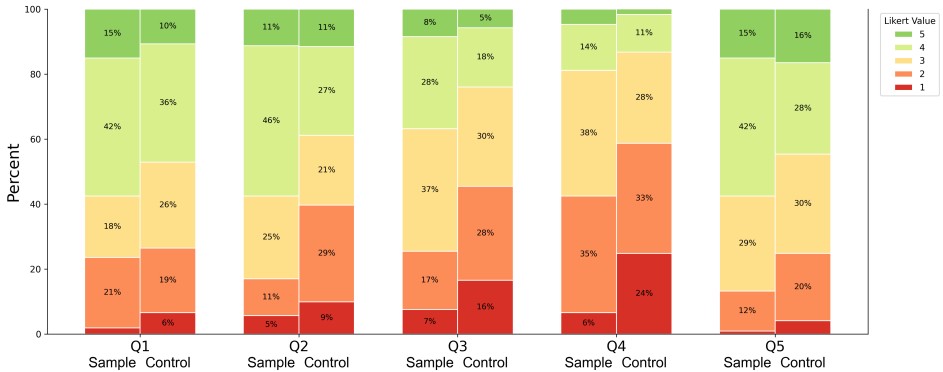

Figure 4: Evaluation of synthetic narratives by human clinical experts.(n=218)

Table 1: Paired t-tests comparing control vs. sample responses across 5 Likert-scale questions. Negative $t$ and $d$ values reflect higher scores in the sample condition. Control and sample show mean ± SD. (*$p < 0.05$, n = 106 pairs).

| Question | Control | Sample | t-stat | p-value | Cohen's $d$ |
|---|---|---|---|---|---|
| Q1 - Clinical Validity | 3.25 ± 1.07 | 3.44 ± 1.05 | −1.367 | 0.175 | −0.139 |
| Q2 - Clinical Timeline Validity | 2.98 ± 1.15 | 3.43 ± 1.04 | −3.049 | 0.003* | −0.310 |
| Q3 - Clinically Actionable | 2.64 ± 1.04 | 3.10 ± 1.04 | −2.964 | 0.004* | −0.301 |
| Q4 - Safely Clinically Actionable | 2.28 ± 0.95 | 2.75 ± 0.92 | −3.593 | <0.001* | −0.365 |
| Q5 - Appropriate Clinical Language | 3.28 ± 1.08 | 3.56 ± 0.94 | −1.973 | 0.051 | −0.201 |

## 6 GENERALIZABILITY AND PIPELINE AGNOSTICISM

While our primary experiments focus on lung cancer—chosen for its prevalence in case report literature, detailed longitudinal trajectories, and heterogeneous treatment paths—the pipeline itself is disease- and language-agnostic. It operates directly on free-text narratives and standardizes outputs via UMLS, integrated into DSPy docstring prompts and JSON output. Ontology grounding ensures that extracted events can be normalized across conditions and datasets.

**Rare Diseases.** We conducted additional experiments on four rare diseases across body systems (Gaucher, Wilson's, Charcot–Marie–Tooth, and Retinitis Pigmentosa), using 50 case reports per

disease (200 total, see Appendix B.2). Despite reduced sample size ($n = 200$ vs. 485) and minimal explicit clinical labels (*e.g.*, metastasis status, subtype), Monopath DAGs consistently improved clustering quality compared to text, with Calinski–Harabasz score gains of +21% (Raw), +24% (PCA), and +101% (UMAP). These findings reinforce the structural advantages of graph-based representations even under constrained conditions, and suggest that larger, labeled cohorts are likely to yield further improvements. While a full-scale evaluation across multiple diseases and languages is beyond the current study, these rare-disease results provide initial evidence of broader extensibility.

## 7 CONCLUSION AND FUTURE WORK

We introduce a data-generation framework that transforms free-text cancer case reports into structured, semantically rich DAGs and then leverages these graphs as blueprints for synthetic patient trajectories. By encoding each narrative as a temporally ordered graph, we capture the rare, nuanced clinical paths that seldom appear in standard EHR corpora and make them available for scalable machine-learning research. We confirm that the graphs faithfully preserve source content, enable meaningful trajectory comparison, and support high-realism synthetic case creation—offering an immediately useful resource for data augmentation, model stress-testing, and downstream clinical reasoning tasks. A central motivation for Monopath DAGs is to enable patient-level causal reasoning. By encoding explicit temporal order and dependencies, they support tasks beyond the reach of bag-of-words, topic models, or sequence encoders. DAGs can be queried for trajectory-aware case matching, edited at the edge level for counterfactual analysis, and used as templates for realistic synthetic cohorts via LLM reconstruction. These capabilities are especially relevant where data are scarce, heterogeneous, or privacy-restricted, moving beyond association mining toward clinically actionable reasoning. Our results underscore the promise of graph-driven synthetic data for studying uncommon disease courses and set the stage for broader deployment across narrative and structured domains.

**Limitations.** This work should be interpreted in light of several constraints. The extraction pipeline is applied and evaluated on a narrow slice of the literature—recent, English-language lung cancer case reports—which simplifies ontology alignment but may not generalize to other specialties, languages, or narrative conventions. The Monopath DAG representation enforces a single dominant clinical thread; as a result, concurrent events, branching trajectories, and cyclic complications are collapsed, omitting potentially important nuances. Although ontology grounding and deterministic topology checks reduce hallucinations, the system still depends on large, general-purpose language models operating in zero-/few-shot mode, and we observe residual errors in negation, temporal ordering, and dosage extraction that are not fully reflected in BERTScore or our limited human evaluation. Realism of the synthetic trajectories is supported only through small-scale expert scoring; we have not yet performed privacy-leakage audits or downstream benchmarking. While the monopath simplification supports clarity and causal interpretability, it inevitably sacrifices expressivity. More complex structures such as multiplex DAGs or hypergraphs could better capture concurrency and branching. We view the present work as establishing a foundation for reliable extraction, with future extensions aimed at richer graph types while preserving directed, acyclic constraints for interpretability.

**Future Work.** We plan to extend our framework beyond to support *Multipath DAGs*—structured patient graphs with controlled branching to capture clinical uncertainty, reversible events, and alternate pathways. To improve extraction fidelity and fill in under-specified clinical states, we will incorporate biomedical LLMs (*e.g.*, ClinicalT5, BioBERT) and graph imputation techniques such as masked node prediction and GNN autoencoders. This will enable more complete and accurate graph representations. Multipath DAGs unlock several downstream applications: trajectory-informed outcome prediction via GNNs, synthetic case generation through branch perturbation, and population-scale DAG repositories for benchmarking and cross-disease comparison. Ultimately, we envision deployment into clinical decision support tools that leverage trajectory-aware graph reasoning to match cases, explore counterfactuals, and inform care planning.

## FUNDING

The authors received no specific funding for this work.

ETHICS STATEMENT

All case reports are peer-reviewed, de-identified, and publicly available via PubMed Central, minimizing re-identification risk; no patient-level EHR data is used.

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

# A   APPENDIX

# A   NOTATIONS AND DEFINITIONS

## A.1   TASK DEFINITION

We focus on the monopath constraint—each node has at most one incoming edge—to establish feasibility of robust, large-scale extraction and to preserve temporal directionality. More expressive graph structures (*e.g.*, multiplex or hypergraphs) can capture concurrent or cyclic phenomena but complicate interpretability. We view them as complementary extensions (see Limitations).

$$\mathcal{D} = \{d_1, d_2, \ldots, d_N\} \tag{1}$$

denotes a corpus of free-text clinical case reports, where each $d_i$ describes a longitudinal patient narrative. For each report $d_i$, our objective is to construct a directed acyclic graph (DAG)

$$G_i = (V_i, E_i) \in \mathcal{G}, \qquad i = 1, \ldots, N, \tag{2}$$

that captures the patient's clinical trajectory in a structured format.

**Graph Representation.** We denote a generic DAG as $G = (V, E)$, where

$$V = \{v_1, \ldots, v_K\}, \qquad E \subseteq V \times V. \tag{3}$$

**Node Semantics.** Each node $v_k \in V$ corresponds to a temporally anchored patient state and is annotated with:

$$
\begin{aligned}
s &: V \to \mathcal{T}, & s(v_k) &= s_k & \text{(free-text content),} \\
a &: V \to \mathcal{A}, & a(v_k) &= a_k & \text{(structured clinical data attributes),} \\
t &: V \to \mathbb{R}_{>0} \cup \{\varnothing\}, & t(v_k) &= t_k & \text{(optional timestamp).}
\end{aligned}
\tag{4}
$$

The attribute set $a_k$ typically includes symptoms, diagnoses, laboratory values, medications, and procedures.

**Edge Semantics.** Each edge $e_{ij} = (v_i \to v_j) \in E$ denotes a transition between two patient states. We define edge annotations as:

$$
\begin{aligned}
\tau &: E \to \mathcal{T}, & \tau(e_{ij}) &= \tau_{ij} & \text{(transition type),} \\
\phi &: E \to 2^{\mathcal{C}}, & \phi(e_{ij}) &= \varphi_{ij} & \text{(triggering clinical entities),}
\end{aligned}
\tag{5}
$$

where $\mathcal{C}$ denotes a standardized clinical ontology [*e.g.*, Unified Medical Language System (UMLS)]. Thus, each $G_i$ yields a machine-readable, temporally ordered abstraction of the unstructured narrative in $d_i$.

**Event Standardization.** To convert free-text narratives into machine-actionable graph elements, all events are normalized into ontology-grounded fields. Sentences are decomposed into atomic facts, and the `NodeClinicalDataExtract` module maps each fact to Unified Medical Language System (UMLS) concepts, yielding a structured `clinical_data` dictionary per node. Edges are similarly represented as typed JSON `transition_event` objects specifying trigger, domain, change type, and ontology-mapped entities ($\phi(e) \subseteq \mathcal{C}$, where $\mathcal{C}$ is the ontology). Narrative strings are retained only as evidence, while ontology grounding enables programmatic querying, aggregation, and embedding-based analysis.

## A.2   NOTATION FOR PATIENT TRAJECTORY GRAPHS

This table summarizes the notation for patient trajectory graphs, including case reports, DAGs, clinical state nodes, and directed transitions. Nodes may include free-text summaries, structured attributes, timestamps, and mapped clinical concepts, while edges capture labeled state transitions. Together, the notation provides a consistent framework for representing patient trajectories.

| Notation | Description |
| --- | --- |
| $D = \{d_1, d_2, \ldots, d_N\}$ | Corpus of free-text clinical case reports |
| $d_i \in D$ | A single case report |
| $G_i = (V_i, E_i)$ | DAG representing the patient trajectory for case $d_i$ |
| $\mathcal{G}$ | Set of all patient trajectory graphs |
| $V = \{v_1, \ldots, v_K\}$ | Set of nodes representing clinical states |
| $E \subseteq V \times V$ | Set of directed edges between states |
| $v_k \in V$ | A single patient state node |
| $s_k \in \mathcal{T}$ | Free-text content of the patient state at $v_k$ |
| $a_k \in \mathcal{A}$ | Structured clinical data attributes at $v_k$ |
| $t_k \in \mathbb{R}^+ \cup \{\emptyset\}$ | Optional timestamp for node $v_k$ |
| $e_{ij} = (v_i \rightarrow v_j) \in E$ | Directed transition from $v_i$ to $v_j$ |
| $\tau_{ij} \in \mathcal{T}$ | Transition type label for edge $e_{ij}$ |
| $\phi_{ij} \subseteq \mathcal{C}$ | Triggering clinical entities mapped to ontology |
| $\mathcal{T}$ | Space of free-text summaries or transition types |
| $\mathcal{A}$ | Space of structured clinical attributes |
| $\mathcal{C}$ | Space of clinical concepts/entities |

# B    ADDITIONAL DATA

## B.1    STRUCTURE AND SEMANTICS OF DAGS

Semantic accuracy is assessed using BERTScore (mean ± standard deviation) across precision, recall, and F1. Topological correctness is evaluated through graph-theoretic metrics: the percentage of acyclic graphs, the number of weakly connected components, and average node-level statistics (node count, edge count, in-degree, and graph density). These metrics quantify both structural validity and interpretability of the generated DAGs.

| Metric | Monopath DAG |
| --- | --- |
| **BERTScore (mean $\pm$ std)** | |
| Precision | $0.790 \pm 0.060$ |
| Recall | $0.807 \pm 0.047$ |
| F1 Score | $0.798 \pm 0.051$ |
| **Topology Validation** | |
| Acyclic | 99.175 % |
| Weakly Connected Component | $1.436 \pm 1.252$ |
| Avg. Node Count | $8.182 \pm 3.139$ |
| Avg. Edge Count | $6.749 \pm 2.567$ |
| Avg. In-Degree | $0.820 \pm 0.108$ |
| Graph Density | $0.137 \pm 0.068$ |

## B.2    RARE DISEASE CLUSTERING OUTPUT

Comparison of Graph vs. Text representations across embeddings. Values shown are (Graph, Text), with percent improvement of Graph over Text. Silhouette score measures how well-separated clusters are, with higher values indicating more coherent and distinct groupings. The Calinski–Harabasz (CH) index evaluates clustering compactness and separation, where larger values reflect tighter, well-defined clusters.

| Dataset | Embedding | Silhouette | | Calinski–Harabasz | |
| --- | --- | --- | --- | --- | --- |
| | | Values | % $\Delta$ | Values | % $\Delta$ |
| Rare Disease | Raw | (0.1340, 0.0978) | +37.0% | (5.12, 4.23) | +21.0% |
| Rare Disease | PCA | (0.2450, 0.1777) | +37.9% | (15.63, 12.56) | +24.4% |
| Rare Disease | UMAP | (0.4845, 0.3727) | +29.9% | (103.25, 51.44) | +100.8% |
| Synthea | Raw | (0.0832, 0.0725) | +14.8% | (3.87, 3.15) | +22.9% |
| Synthea | PCA | (0.1904, 0.1627) | +17.0% | (8.55, 6.97) | +22.7% |
| Synthea | UMAP | (0.3328, 0.2301) | +44.7% | (5.11, 4.62) | +10.6% |

## C  OUTPUT OVERVIEW

### C.1  NODE AND EDGE ATTRIBUTES

The following table lists the structured attributes used in our graph representation of patient trajectories. Nodes represent individual clinical states, while edges denote transitions between those states. Not every field is present in every case. Only available information from the narrative is populated.

| Node Attributes | Edge Attributes |
| --- | --- |
| **Node ID:** unique identifier (e.g., "A", "B") in order of appearance. | **Edge ID:** concatenation of source and target node IDs (e.g., "A→B"). |
| **Step Index:** index in the sequence. | **Branch Indicator:** true if edge starts a side branch. |
| **Narrative Content:** descriptive text of the state. | **Narrative Content:** description of the event or change. |
| **Timestamp:** extracted if explicitly mentioned. | **Transition Event:** structured encoding with: |
| **Clinical Data:** structured key-value fields (e.g., vitals, labs, medications, imaging, HPI, etc.). | `Trigger Type`, `Trigger Entities`, `Change Type`, `Affected Domain` |

### C.2  NODE OUTPUT EXAMPLE

The following is an example of structured content for a node extracted from a patient case report. It includes narrative summary, structured clinical data (such as HPI and social history), and concept codes.

//

---

**Node Details: `graph_001`**

- **id:** N1
- **label:** Step 1
- **customData:**
  - **node_id:** A
  - **node_step_index:** 0
  - **content:** 44-year-old male presented with right-sided chest pain, dry cough, on-off fever, and hematuria for 2 months. Patient has no history of Antitubercular treatment (ATT) intake and was a tobacco chewer for >20 years. Physical examination revealed decreased air entry on the right side of the lung.
  - **clinical_data:**
  - **HPI:**
    - **summary:** right-sided chest pain, dry cough, on-off fever, and hematuria for 2 months
    - **duration:** 2 months
    - **associated_symptoms:**
    - C0008031
    - C0010200
    - C0015967
    - C0019062
  - **social_history:**
    - **category:** tobacco
    - **status:** current
    - **description:** tobacco chewer for ¿20 years
- **custom_id:** graph_001_N0

---

### C.3  EDGE OUTPUT EXAMPLE

The following is an example of a structured edge output from a case, representing a transition between two clinical states. It includes both narrative summary and structured transition fields.

//

> **Edge Details: `graph_001`**
>
> - **from:** N1
> - **to:** N2
> - **data:**
>   - **edge_id:** A_to_B
>   - **branch_flag:** true
>   - **content:** Initial presentation and subsequent imaging
>   - **transition_event:**
>     - **trigger_type:** imaging
>     - **trigger_entities:**
>     - **change_type:** addition
>     - **target_domain:** imaging
> - **custom_id:** graph_001_N1_N2
> - **id:** 413a3142-63b0-4eeb-9830-26011a516e1a

## C.4 DAG CONSTRUCTION ALGORITHM

To illustrate how free-text case reports are systematically transformed into structured trajectory graphs, we provide the procedure below. The algorithm outlines the steps for segmenting narratives, constructing nodes and edges, extracting clinical data, and serializing the resulting Monopath DAG representation.

---

**Algorithm 1** DSPy-Guided Monopath DAG Construction

---

**Require:** Free-text case report $d_i$
1: $p_{1:k} \leftarrow \text{segment}(d_i, \text{window} = 10 \text{ sentences})$
2: $T_i \leftarrow \text{PATIENTTIMELINE}(p_{1:k})$                                           $\triangleright$ LLM signature
3: **for** block $c_j$ in $\text{chunk}(T_i, 4 \text{ sentences})$ **do**
4:     $v_j \leftarrow \text{NODECONSTRUCT}(c_j, \text{mem} = \{v_{<j}\})$
5:     $s^{\text{atom}} \leftarrow \text{DECOMPOSETOATOMICSENTENCES}(v_j.s)$
6:     $a_j \leftarrow \text{NODECLINICALDATAEXTRACT}(s^{\text{atom}})$
7:     $\{v_{<j}\} \leftarrow \text{MERGEIFDUPLICATE}(v_j, \{v_{<j}\})$
8: **end for**
9: **for** $i = 1$ **to** $|V| - 1$ **do**
10:     $e_{i \rightarrow i+1} \leftarrow \text{EDGECONSTRUCT}(v_i, v_{i+1})$
11:     $b_i \leftarrow \text{BRANCHCLASSIFY}(e_{i \rightarrow i+1})$
12: **end for**
13: $G_i \leftarrow \text{SERIALIZEASJSON}(V, E, \text{UMLS metadata})$
14: **return** $G_i$

---

## C.5 BRANCH ALGORITHM

We implement a post-processing procedure to collapse redundant branches and ensure consistent step indexing across trajectories. Algorithm 2 outlines the semantic branch collapse and step index propagation routine, which merges nested branches and propagates indices from root nodes through the graph.

---

**Algorithm 2** Semantic Branch Collapse and Step Index Propagation

---

1: **procedure** PROPAGATEANDCOLLAPSE($G$)
2:     $visited \leftarrow \emptyset$
3:     **for all** $node \in G.nodes$ **do**
4:         **if** ISBRANCHORIGIN($node$) **and not** ISINNERMOSTBRANCH($node$, $G$) **then**
5:             COLLAPSEBRANCHTOPARENT($node$, $G$)
6:         **end if**
7:     **end for**
8:     $roots \leftarrow \{n \in G.nodes \mid n.parents = \emptyset\}$
9:     **for all** $root \in roots$ **do**
10:         ASSIGNSTEPINDEX($root$, $G$, $0$, $visited$)
11:     **end for**
12: **end procedure**
13: **function** ISBRANCHORIGIN($node$)
14:     **return** $node.\texttt{metadata}[\text{``branch\_origin''}] = \text{True}$
15: **end function**
16: **function** ISINNERMOSTBRANCH($node$, $G$)
17:     $stack \leftarrow [node]$
18:     **while** $stack \neq \emptyset$ **do**
19:         $current \leftarrow \text{POP}(stack)$
20:         **if** ISBRANCHORIGIN($current$) **and** $current \neq node$ **then**
21:             **return False**
22:         **end if**
23:         **for all** $child \in current.children$ **do**
24:             PUSH($stack$, $G[child]$)
25:         **end for**
26:     **end while**
27:     **return True**
28: **end function**

---

# D    HUMAN EVALUATION

## D.1    HUMAN EVALUATION SCHEMATIC

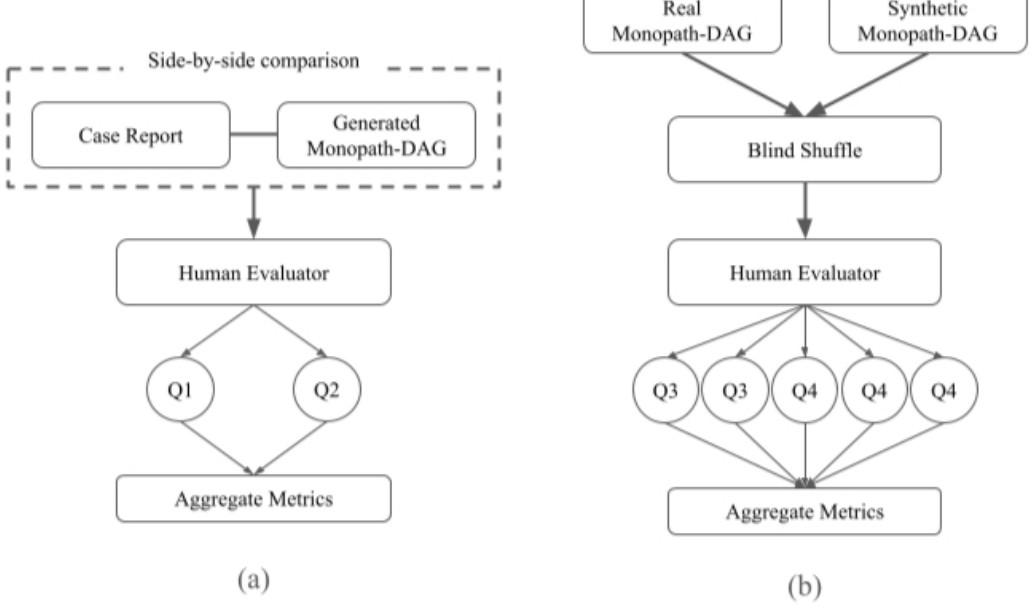

Figure 5: Human evaluation schematics for (a) fidelity and (b) synthetic graph assessments.

## D.2    FIDELITY EVALUATION

To assess whether generated Monopath-DAGs faithfully represent the underlying case reports, we implemented a two-part fidelity evaluation. First, we reconstructed narratives from graphs and compared them to original reports with BERTScore (ClinicalBERT embeddings). This provided automated precision, recall, and F1 measures of semantic similarity. Second, human annotators performed side-by-side comparisons of case reports and corresponding graphs, answering the following questions:

- **Q1:** Are the nodes and edges in the correct order? [Yes / No]
- **Q2:** Is the content of the graph accurate to the case report? [Yes / No]

Aggregate metrics from these assessments quantify structural and semantic fidelity (Figure 5a).

## D.3    CLINICAL EVALUATION

To move beyond correctness, we conducted a clinician-centered evaluation focused on decision-making relevance. Real and synthetic Monopath-DAGs were blind-shuffled and presented to practicing physicians, who rated each on five targeted criteria:

- **Q1 (Clinical validity):** Are the case actions and insights medically sound?
- **Q2 (Timeline validity):** Does the case progression follow a clinically appropriate order?
- **Q3 (Clinically actionable):** Is the case specific and granular enough to guide the next clinical decision?
- **Q4 (Safely clinically actionable):** Is the information sufficient to make a safe decision with confidence?
- **Q5 (Appropriate language):** Does the narrative use medical language appropriately, avoiding generic or ambiguous phrasing?

These criteria capture reasoning dimensions absent from standard benchmarks but essential for clinical impact. Aggregate metrics from these assessments provide a structured view of graph quality in real-world settings.

# E PROMPT DESIGN

## E.1 DAG PRIMER

Docstring used to prime the LLM for DAG extraction.

---

**Docstring: `dag_primer`**

```
You are an assistant that converts clinical case narratives into
dynamic Directed Acyclic Graphs (DAGs).

Each DAG consists of:
- Nodes = snapshots of the patient's state.
- Edges = transitions between those states.

Terminology guidance:
- Use UMLS-standard concepts when possible for consistency and
interoperability.
- If a concept isn't covered by UMLS, use clear, logical labeling.

Text extraction guidance:
- When looking at the case report input, ignore the references,
background, conclusions etc.  sections
- We only want to extract on content relating to the specific
patient discussed in the case report
```

## E.2 NODE INSTRUCTIONS

Instructions used to guide LLM-based node construction.

---

**Docstring: `node_instructions`**

```
You are given a clinical case report.  Your task is to extract a
sequence of nodes representing the patient's evolving clinical
state.

Guidelines:
- Create one node per clinically meaningful state.
- Combine co-occurring labs/imaging into the same node.
- Use separate nodes for clearly sequential or distinct events.
- Do not return anything outside the list format.  Should be in JSON
compatible style.
- Keep imaging content packaged in one node if no clear temporal
change is indicated.
- Keep pathology / histology content packaged in one node if no
clear temporal change is indicated.
- node_memory is a running memory that updates as we add new nodes;
use it to preserve context, merge overlapping details, and avoid
redundant or stale states.

Output format:
Return a list of node dictionaries, in this order from top to
bottom, each with:
- node_id (In ascending alphabetical order, e.g., "A", "B", "C")
- node_step_index (integer for order)
- content (exhaustive clinical content, include all relevant details
for the given node)
- timestamp (optional, ISO8601)

Example:
{
"node_id":  "A",
"node_step_index":  0,
"content":  "The patient presented with bilateral painless
testicular masses."
}
```

---

## E.3 EDGE INSTRUCTIONS

Instructions used to guide LLM-based edge construction.

---

**Docstring: `edge_instructions`**

```
Each edge represents a change from one node to another.  There
should be one edge in between every pair of adjacent nodes.

Guidelines for edges:
- Create edges only when there is a clear clinical progression or
change between nodes.
- Maintain narrative or logical order | edges should flow from
earlier to later events.
- Combine co-occurring findings into the same node, not across
multiple edges.

Output format:
Return a list of edge dictionaries, in this order, each with:
- edge_id:  Unique identifier (Use format "node_id"_to_"node_id",
such that the first "node_id" is the upstream node and the second
"node_id" is the downstream node bounding the edge)
- branch_flag:  Boolean if this starts a side branch, default = True
- content:  Exhuastive clinical content, include all relevant
details for the given node

Optional structured field for edge-level transitions:
transition_event = {
    "trigger_type":  "procedure | lab_change | medication_change |
symptom_onset | interpretation | spontaneous",
    "trigger_entities":  ["UMLS_CUI_1", "UMLS_CUI_2"],  # e.g., C0025598
= Metformin, C0011581 = Chest Pain
    "change_type":  "addition | discontinuation | escalation |
deescalation | reinterpretation | resolution | progression | other",
    "target_domain":  "medication | symptom | diagnosis | lab |
imaging | procedure | functional_status | vital_sign",
    "timestamp":  "ISO 8601 datetime (e.g., 2025-03-01T10:00:00Z),
only include if explicitly given and can be converted to datetime"
}
```

---

### E.4 Node clinical data instructions

Instructions for extracting structured clinical data per node.

---

**Docstring: `node_clinical_data_instructions`**

```
Each node includes an optional structured field clinical_data,
formatted as a dictionary with categories mapping to lists of
dictionaries.  Values should only be included if matched to the
UMLS Metathesaurus; otherwise, omit and do not print/store it.

clinical_data = {
  "medications":  [{
    "drug":  "UMLS_CUI or string",
    "dosage":  "string",
    "frequency":  "string",
    "modality":  "oral | IV | IM | subcutaneous | transdermal |
inhaled | other",
    "start_date":  "ISO8601",
    "end_date":  "ISO8601 or null",
    "indication":  "UMLS_CUI or string"
  }],
  "vitals":  [{...}],
  "labs":  [{...}],
  "imaging":  [{...}],
  "procedures":  [{...}],
  "HPI":  [{...}],
  "ROS":  [{...}],
  "functional_status":  [{...}],
  "mental_status":  [{...}],
  "social_history":  [{...}],
  "allergies":  [{...}],
  "diagnoses":  [{...}]
}
```

---

## E.5 BRANCH INSTRUCTIONS

Instructions for detecting and labeling branching transitions in the patient trajectory.

```
Docstring: branch_instructions

Branches arise when physiologic changes or complications aren't part
of the main pathway but impact patient states.  Specifically, we are
thinking of ephemeral changes.

Mark side branches clearly:
- Edge initiating a new branch:  branch_flag = True
```

# F  ADDITIONAL RELATED WORK

## F.1  EXPANSION BEYOND EXISTING EXTRACTION METHODS

We situate Monopath DAGs within prior efforts on case report extraction and patient-journey modeling. The below table summarizes representative approaches, contrasting pipelines, outputs, and downstream utility. Whereas earlier resources focus on entity spans, citation heuristics, or free-text knowledge graphs, our method provides temporally ordered, ontology-grounded DAGs that directly capture the structure of patient trajectories.

| Work | Extraction Pipeline | Output | Utility |
|------|--------------------|--------|---------|
| **Monopath DAGs** | LLM + DSPy pipeline | UMLS-grounded DAG per patient | Supports trajectory retrieval, counterfactual edits, synthetic cohorts; clinically validated |
| **PMC-Patients** (Zhao, 2023) | Regex & heuristics; citation links | Paragraph + triples | Retrieval benchmark only; no temporal or causal |
| **CaseReportBench** (Zhang, 2025) | Human BIO annotation | Entity spans | High-quality labels; no temporal or relation structure |
| **Patient-Journey KGs** (Al Khatib, 2025) | Prompt-based LLM; node merging | Multi-relational KG per patient | Structural compliance evaluation; no clinical validation |

**Comparison with PMC-Patients.**    While both our work and Zhao et al. (PMC-Patients) leverage case reports, the approaches differ fundamentally.

*Scope:* PMC-Patients is designed as a retrieval benchmark using free-text patient summaries and citation links, whereas our work extracts structured representations and models causal dependencies across events.
*Granularity:* PMC-Patients preserves paragraphs, but Monopath DAGs break narratives into machine-readable node–edge pairs, annotated with timestamps and UMLS terms, enabling clustering, simulation, and downstream reasoning.
*Evaluation:* Zhao et al. report retrieval performance; we instead validate semantic fidelity (BERTScore), demonstrate improved trajectory clustering (Calinski–Harabasz), and conduct blinded physician assessments of synthetic realism.
*Use-case:* By encoding causal flow, DAGs enable applications beyond retrieval, including trajectory-aware search, counterfactual simulation, and synthetic cohort generation.

## G   COMPUTATIONAL RESOURCES

The pipeline is implemented in Python and runs on standard research hardware. Experiments used a single NVIDIA A100 GPU (40GB) with 64GB RAM, though CPU-only execution is possible with longer runtimes. End-to-end processing takes about 2–3 minutes per case report, making the approach accessible to academic and clinical groups.

