# OpenReview forum: "Monopath DAGs: Structuring Patient Trajectories from Clinical Case Reports"
_ICLR.cc/2026/Conference — ICLR 2026 Conference Withdrawn Submission_

### Official Review · Reviewer_TAzr · 2025-10-29

**Soundness:** 1
**Presentation:** 1
**Contribution:** 1
**Rating:** 2
**Confidence:** 4

**Summary:**

The authors present an approach to summarizing clinical case reports as graphs, based on a pipeline of variously prompted LLMs. The approach is primarily evaluated on a set of lung cancer case reports.

**Strengths:**

* The hypothesis that conversion of reports to graphs can result in more useful LLM summaries is worth exploring.
* Contains a review of graphs by four practicing physicians
* Demonstrates that extracted graphs degrade (over the original text) the event order by only 13% and the event description by only 6% .

**Weaknesses:**

Major:
* The paper does not test its central assumption, that a graph-based summary of a case report is more useful than the original case report. Without some clear evaluation of the utility of the graphs, it's hard to justify the rest of the paper that focuses on the construction of such graphs.
* Critical components of the pipeline are not defined: PatientTimeline, NodeConstruct, DecomposeToAtomicSentences, NodeClinicalDataExtract, edgeConstruct, branchClassify, LLMReconstructor. I'm guessing they're defined as prompt+LLM, but no information is given in the main text.
* The semantic fidelity evaluation as described is problematic, as an LLM is given the graph and "prompted to generate a coherent textual summary that mirrors the structure and content of the original case", implying that the LLM sees not only the graph, but the original case it is supposed to mirror. The LLM could easily mirror the original structure and content by ignoring the graph. Thus, the evaluation needs to be changed to not provide the original case report when asking an LLM to generate a case report from the graph.
* It's not clear what counts as a "real patient graph" and a "synthetic patient graph". The only thing discussed under "synthetic generation" are cases that are converted into snippets then back into prose, and cases that are converted into graphs and back into prose. Is one of those supposed to be "real" and one "synthetic"? They both sound real to me since they're both generated from the real case report. Moreover, why are these two things being compared, rather than comparing the original case report vs. the graph-based summarization of the case report?
* It is not clear what the clustering is over. Nodes? Edges? Entire graphs? And what is the goal of the clustering? That is not specified. The motivation for clustering as an evaluation is also not explained. Does a good clustering score correlation with some downstream usability metric?
* The metastatic distribution analysis of the clusters isn't well justified. If the goal is to detect metastatis, then compare the original case report vs. the graph-based summary of the case report as input to a classifier, and show which one yields better predictive power.

Minor:
* There is no figure in the main text showing the reader what a clinical case report looks like, nor what a monopath directed acyclic graph for that report looks like. This is important for quickly familiarizing the reader with the input and output of the proposed task. This could be addressed by a new figure in the main text that couples the example output node and edge from the appendix with other nodes and edges in the graph along with whatever input those nodes and edges were extracted from.
* The introduction says that "case reports have remained largely inaccessible", though they have been the subject of NLP research for many years: https://aclanthology.org/N18-1140/, https://aclanthology.org/W19-5029/, https://aclanthology.org/2020.lrec-1.553/, etc. Some rephrasing is needed to clarify what the authors mean by "inaccessible" here.
* Some contextualization of the BERTScore scores is needed. What does a pair with a score of 0.876 look like? What about a pair with score of 0.676? This is needed to set the reader's understanding of meaning of the BERTScore range. This is especially necessary given the possibly overly broad claim that the proposed approach "reliably reproduces the core clinical information and narrative structure across diverse cases" when BERTScore doesn't directly evaluate clinical information or narrative structure.
* Figure 2's caption mentions (a), (b), (c), and (d), but there are no such labels in the figure.

**Questions:**

See above.

---

### Official Review · Reviewer_XnTS · 2025-10-30

**Soundness:** 2
**Presentation:** 3
**Contribution:** 2
**Rating:** 2
**Confidence:** 4

**Summary:**

This paper investigates the use of narrative case reports for structured patient trajectory modeling. The authors propose a modular framework that converts free-text case reports into Monopath Directed Acyclic Graphs (DAGs), enabling temporally ordered and semantically coherent representations of patient progressions. The framework is evaluated through semantic fidelity and clustering analyses on both lung cancer and rare disease case reports, demonstrating the framework’s potential to extract meaningful and interpretable structures from unstructured clinical narratives.

**Strengths:**

**S1.** The paper addresses a practically important and clinically meaningful problem by exploring how narrative case reports—an underutilized yet rich source of clinical information—can be leveraged for machine learning applications in healthcare.

**S2.** The proposed framework integrates large language models (LLMs), rule-based extraction, and clinical ontology alignment to identify key entities, events, and transitions within free-text narratives. This integration enables the construction of Monopath DAGs that preserve both temporal clarity and semantic precision, providing a structured representation of patient trajectories.

**S3.** The framework is evaluated on a curated corpus of lung cancer case reports and extended to four rare diseases across the body, demonstrating its generalizability in modeling diverse clinical scenarios.

**Weaknesses:**

**W1.** The proposed framework for patient trajectory modeling, i.e., the DAG generation pipeline described in Section 3, primarily integrates established techniques, including LLMs, rule-based extraction, and clinical ontology alignment. As a result, the methodological novelty and technical depth of the work appear somewhat limited. Moreover, the connection between the proposed framework and the core research themes of the conference could be more clearly articulated to highlight its relevance and potential contributions to the targeted audience.

**W2.** The relationships between the proposed framework and existing studies are not sufficiently clarified. While Section 2 outlines three related research lines that align with the functionalities of the proposed framework, it remains unclear which prior methods serve as direct baselines for comparison. Consequently, the specific advantages and added value of the proposal within each research line are difficult to discern. A clearer positioning of the work relative to existing methods would strengthen the paper’s contributions and contextual grounding.

**W3.** The experimental evaluation appears to serve primarily as a feasibility study rather than a comprehensive validation. In the absence of appropriate baseline methods for comparison, the effectiveness of the proposed framework remains insufficiently substantiated. While Appendix B.2 includes a brief comparison with Synthea (without corresponding discussions), Synthea itself does not represent a recent or technically competitive counterpart. It would be necessary to consider incorporating the methods mentioned in Appendix F.1 as comparative baselines, which could provide a more rigorous and convincing empirical assessment of the proposed framework’s advantages.

**W4.** The paper suggests that the proposed framework can be extended to support additional functionalities, such as patient-level causal reasoning and trajectory-aware case matching. While these directions are promising, the claims would be more convincing if supported by preliminary experimental evidence or illustrative case studies.

**W5.** The overall presentation of the paper could be improved to enhance readability and coherence:

* The methodology section would benefit from the inclusion of a running example to help readers better follow the key steps and techniques of the proposed framework.

* In Section 3.2, additional details and background on DSPy would improve clarity, particularly for readers less familiar with this component and its integration within the framework.

**Questions:**

Beyond W1-W5, I have the following questions for clarification:

**Q1.** How are the Monopath DAGs generated by the proposed pipeline intended to complement existing electronic health records (EHRs)? In cases where discrepancies or contradictions arise between the DAG-derived trajectories and the structured EHR data, how would such inconsistencies be identified and reconciled?

**Q2.** It appears that “Cluster graph-2” referenced in the text is not displayed in Figure 3(a).

---

### Official Review · Reviewer_rW5Q · 2025-10-31

**Soundness:** 1
**Presentation:** 1
**Contribution:** 3
**Rating:** 2
**Confidence:** 4

**Summary:**

The authors propose a method to structure patient trajectories from clinical case reports. They use LLMs to transform a set of cancer case reports to monopath DAGs, a type of DAG that is temporarily ordered and reflects the sequence of clinical decisions and observations described in the case report. These structured patient trajectories form a machine-readable version of such case reports, enabling training of CDSS on rare disease and modeling causal reasoning, as well as forming a template for synthetic data generation to preserve privacy of sensitive reports.

**Strengths:**

Structuring case reports into machine-readable formats is a very relevant contribution and I can see such a dataset being used for multiple downstream tasks. As stated by the authors, such a resource is especially valuable to train machine learning models which are more aligned with real-world clinical reasoning. Generating a synthetic version of a real report through the intermediate step of a monopath DAG also holds great potential for privacy-preserving unstructured data generation. The expert review is a very valuable step for data validation and seems to be conducted with care and rigor.

**Weaknesses:**

The narrative of the paper is quite unclear and seems to shift throughout the paper, so it is hard for me to identify what the central claim of the paper is. While it is clear that the authors aim to build DAGs to represent patient trajectories, it is unclear what these DAGs should be used for, and therefore unclear what the adequate experiments should be to assess utility of the dataset.

- From the introduction and related work, I interpreted the main goal to be the following: to build machine-readable structured DAGs from unstructured case reports to enable clinical decision support on rare diseases (as high-quality structured datasets of EHRs are lacking for such cases). However, the case reports considered by the authors are in the field of cancer, which is certainly not a rare disease, and the small set of experiments on rare diseases in Section 6 does not make up for this, as an evaluation study with clinical experts is missing there. A parallel goal of the dataset (according to the introduction) is to enable the training of machine learning models which are more aligned with real-world clinical reasoning processes, but later on in the paper they do not assess whether their dataset fills this gap or demonstrate how it might be used to fulfill this goal. As stated by the authors in Section 5 “in this study, graphs are used as structured intermediates to support generation and retrieval, rather than as interfaces intended for direct clinical decision-making”, which seems to contradict the central claim posited in the introduction.

- In the remainder of the paper, including the conclusion, the main goal seems to shift towards using these monopath DAGs to build convincing synthetic patient trajectories. Apart from general validation of the DAGs (Section 4.1 and 4.2, which I deem to be sound enough), a large part of the experiments and discussion is focused on a synthetic trajectory experiment, and the conclusion seems to promote this as a core contribution. While this is an interesting research avenue, this is not the main claim that is advertised in the introduction, causing the reader to feel misled and confused. Furthermore, the setup of the experiment with synthetic trajectories was entirely unclear to me, rendering a large part of the experiments hard to evaluate for soundness and thereby unconvincing.

- Both are valid and valuable research directions, but my main issue is that a large part of the experiments support only the second goal (while being poorly presented), while the introduction seems to present the first idea as a key goal.



There are unsupported claims in Section 4.2 and Section 5, regarding the clustering experiment.

- Here, the authors are “clustering patient trajectories”, though they aggregate node embeddings with mean pooling, thereby losing the ordering and trajectory component of the DAGs.

- Furthermore, the authors state that “graph-based embeddings more distinctly stratify patients by metastasis status compared to text-based representations”, this claim does not seem to be supported by Figure 3, where both the graph-based and text-based results contain clusters with mixed metastasis labels.



The presentation of the paper should be improved. Many explanations in Section 3 and Section 4 come across as vague and incomplete.

- As discussed in the first point, the paper would greatly benefit from a clearer narrative, ensuring that the introduction aligns with the experiments and the core contributions which are put forward in the conclusion, aspects which currently seem misaligned.

- Section 3 contains many technical details about the DSPy-guided DAG generation, but I think it would be more instructive to provide a general overview and intuition behind the extraction pipeline, and leave the technical setup for the appendix. The authors seem to assume that the reader is already familiar with DSPy. If the reader is not familiar, it is unclear which parts of the extraction pipeline are novel modules designed by the authors and which are standard practice in DSPy.

- I found 4.3 very hard to understand, making it impossible for me to check the soundness of these experiments. The terms “real” vs. “synthetic”, and “control” vs. “sample” cases are used interchangeably, yet it is not clear how they map onto each other. A “clinically meaningful synthetic data generator” is listed as a key contribution in the introduction, yet Section 4.3 glosses over how such a synthetic data generator would work, and this is not clarified further in the Appendix either.

**Questions:**

- Is your goal to build machine-readable structured patient trajectories as a novel training resource for clinical decision support methods? Or is your goal to use the generated DAGs to generate more convincing synthetic reports? The introduction suggests the former, while the experiments and conclusion suggest the latter.

- Could you please clarify what control cases and sample cases are in Section 4.3? Which of these are synthetic and which of these are real? How are control cases reconstructed from the original HTML report, and what DSPy prompts are used for this? Why are sample cases generated with depth-first search and then reconstructed? What exactly do you want to prove with the experiment described in Section 4.3?

- What is the background of the human clinical experts, i.e. their specialization? Are they specialized in cancer or are they general practitioners?

- Figure 2 is missing a, b, c, d labels, while these are referenced in the caption.

- The steps visualized in Figure 1 should align better with the terms used in Section 3.2, to improve understanding and reproducibility. That being said, I found algorithm C.4 in the appendix to be more instructive than Figure 1, so consider swapping these around.

- I assume Appendix E contains the specific prompts used to extract nodes and edges, though this appendix is not referenced anywhere in the main text, and the names of these routines are not mentioned in Section 3.2, making the DAG generation method hard to understand and reproduce.

- I feel that it would be valuable to the reader if they could see an example of a (section of a) case report, alongside the extracted monopath DAG. I expected to see such a thing in Figure 2a instead of a general example of a monopath DAG. Minimal adaptations to this figure, like adding annotations for clinical events, would greatly improve the utility of this figure.

- Figure 3a shows graph-0 and graph-1, while the text talks about graph-0, graph-1 and graph-2. Which one is correct?

- Consider using subsections in Section 5, which align with Sections 4.1, 4.2 and 4.3 for increased readability. Or, consider merging Section 4 and 5 altogether, ensuring that the results for each experiment (Section 5) follow directly after the experiment description (Section 4)

- What is the inter-expert agreement in Figure 2d? Does Q1 measure order and Q2 measure accuracy (as implied by the caption of Figure 2), or is it the other way around (as implied by the evaluation criteria at the bottom of p.6)?

- In section 7, you mention “DAGs can be .. used as templates for realistic synthetic cohorts via LLM reconstruction”. Why would you need synthetic versions of case reports, which are already freely available and anonymized with regards to patient identity? Why would you structure a case report as a DAG before reconstructing it with an LLM, when you could just use the original case report directly?

---

### Official Review · Reviewer_NNAP · 2025-11-01

**Soundness:** 2
**Presentation:** 1
**Contribution:** 2
**Rating:** 4
**Confidence:** 4

**Summary:**

The paper presents Monopath DAGs, a framework that converts clinical case reports into structured, time-ordered patient trajectory graphs. Evaluations show the approach preserves clinical fidelity and improves clustering and synthetic data generation over text-based methods.

**Strengths:**

The strengths are:

1. Comprehensive and carefully designed graph construction from clinical notes that can help with synthetic patient data generation
2. Various ways to evaluate the usefulness of the constructed graphs, including clinical notes reconstruction, clustering, and synthetic data generation.

**Weaknesses:**

The following are weaknesses of the paper in my opinion:

1. Writing clarity needs a lot of improvement. Section 3 on the graph extraction process is VERY hard to follow with heavy, ambiguous, sometimes undefined jargons/terms/notations (e.g., p_k is not defined in the main paper; what is a “timeline generator” and the output of it? what do “category-wise” and “atomic-level” mean? what does “BootstrapFewShot” mean? and a lot more).
    * It might also be helpful for the authors to provide the necessary medical background knowledge (maybe via a motivating/toy example) earlier in the paper, so that readers would be familiar with different concepts of patient trajectories when referring to them in Section 3 and later parts.
    * I would suggest that the authors proofread the paper carefully.
    * Line 195, the hyperparameters are chosen “heuristically”? What do you mean by it?
2. Section 4.1. The expert evaluation on the graph validity seems to be overly simple, with only two binary (yes/no) questions.
3. In Section 4.3, the baseline synthetic data generation process seems to be very heuristic-driven. Is it according to some existing literature? If not, it makes the comparison between the baseline and their proposed graph-based generation invalid, as we don’t know whether the baseline is even reasonable. For the synthetic patient data generation methods listed in Section 2, are any of those applicable to become a baseline here?

Minor:

4. Figure quality can be improved. Both the labels of the x/y axes used in the figures, and the resolution of them (many of the graphs are blurred).

**Questions:**

1. The authors claim graphs, schema, and code are released, but I couldn’t find any supplementary material or anonymized code link in the PDF. Did I miss anything?
2. Maybe I missed this in the draft, but is there a pre-defined “vocabulary” for the nodes (patient status) and edges (disease progression)? If not, how do you know they are clinically meaningful? If yes, how is it used in the graph generation process, and is the process of constructing this vocabulary scalable if you were to look at a different disease type? Besides, what are some potential challenges when dealing with rare diseases, e.g., the “vocabulary” could be sparse and keeps evolving over time as people build understanding towards those rare cases?
3. In Section 4.1, when you use precision and recall to evaluate the fidelity of the graph, how should we interpret high versus low precision or recall? Can you elaborate on the trade-off between precision and recall?
    * Besides, for a given original case report, can you also check if the most similar reconstructed narrative actually corresponds to the one that is reconstructed from the same original case report? You can calculate a simple “match rate” as part of the evaluation metrics.
4. In Section 4.2, how are token-level embeddings used in the clustering?
5. For the prompt associated with different tasks listed in the Appendix, can you describe the process of coming up with them and how you end up using them? Any refinement steps as you iterate the prompt versions that’re worth noting?

**Details Of Ethics Concerns:**

n/a.

---

### Note · Authors · 2025-12-03

I have read and agree with the venue's withdrawal policy on behalf of myself and my co-authors.